# Low Profile Dual-Band Polarization Conversion Metasurface with Omnidirectional Polarization

**DOI:** 10.3390/ma16124347

**Published:** 2023-06-13

**Authors:** Jun-Jie Zhang, Wei-Xi Xu, Yu-Tong Zhao, Han-Yu Xie, Hao-Ran Zu, Bian Wu

**Affiliations:** 1National Key Laboratory of Antennas and Microwave Technology, Xidian University, Xi’an 710071, China; msn_zjunjie@163.com (J.-J.Z.); xuweixi@stu.xidian.edu.cn (W.-X.X.); xiehanyu@stu.xidian.edu.cn (H.-Y.X.); hrzu@stu.xidian.edu.cn (H.-R.Z.); 2Southwest China Research Institute of Electronic Equipment, Chengdu 610036, China

**Keywords:** polarization conversion, dual-band, low profile, omnidirectional polarization

## Abstract

In this work, a dual-band transmissive polarization conversion metasurface (PCM), with omnidirectional polarization and low profile, is proposed. The periodic unit of the PCM is composed of three metal layers separated by two substrates. The upper patch layer of the metasurface is the patch-receiving antenna, while the bottom layer is the patch-transmitting antenna. Both antennas are arranged in an orthogonal way so that the cross-polarization conversion can be realized. The equivalent circuit analysis, structure design, and experimental demonstration are conducted in detail, the polarization conversion rate (PCR) is greater than 90% within two frequency bands of 4.58–4.69 GHz and 5.33–5.41 GHz, and the PCR at two center operating frequencies of 4.64 GHz and 5.37 GHz is as high as 95%, with a thickness of only 0.062$\lambda_{L}$, where $\lambda_{L}$ is the free space wavelength at the lowest operating frequency. The PCM can realize a cross-polarization conversion, when the incident linearly polarized wave at an arbitrary polarization azimuth, which indicates that it has the characteristics of omnidirectional polarization.

## 1. Introduction

Polarization converters are devices that can control the polarization direction of electromagnetic waves. They are mainly used in optics to precisely modulate the polarization of light [1,2,3,4,5]; in the field of ultrasound they can be used for imaging [6], and in the microwave field, they can be used where the polarization direction of the electromagnetic wave is particularly demanding, such as in satellite communications [7,8,9], electromagnetic stealth [10], navigation [11], and radar [12,13].

Generally, polarization converters can be classified into linear polarization converters [14,15,16], linear-to-circular polarization converters [17,18,19], and circular polarization converters [20,21] according to their uses. A linear polarization converter can rotate the polarization azimuth of an incident linearly polarized wave by a certain angle, such that the polarization mode of the incident wave is consistent with that of the receiver antenna at the terminal, and the energy loss due to polarization mismatch can be reduced. In wireless communication systems, the polarization conversion of incident waves can be accomplished by loading polarization converters without mechanical rotation of the system, or by switching the antenna configuration and feeding pattern of the system. The existing linear polarization converters are designed according to the fixed polarization azimuth of the incident wave [22,23] which makes the efficiency of the polarization converters unstable when the polarization azimuth of the incident wave is unknown. However, if the polarization azimuth difference between the transmitting antenna and the receiving antenna is up to 90°, the transmission coefficient between the two antennas will be zero, which limits the application of the existing polarization converters to a large extent.

This work is organized as follows. In Section 2, the design principles of the PCM are explained in detail. The performance of the proposed PCM is demonstrated by both simulations and measurements, and it is presented and analyzed in depth by means of field analysis and equivalent circuit theory. In Section 3, the prototype PCM is fabricated and measured, and the measured and simulated results are compared and discussed. Finally, the conclusion is expounded and presented in Section 4.

## 2. Design and Analysis

In this section, the geometrical structure of the proposed omnidirectional PCM will first be presented in detail. An equivalent circuit model is proposed to explain the contributions of different parts of the proposed PCM. The comparison and discussion between the full-wave simulation results and equivalent circuit analysis are presented. Then, the distributions of electric field and current are investigated and analyzed for further verification.

### 2.1. Design of PCM Element

Figure 1 shows the unit cell of the proposed PCM, where all the yellow parts are copper foils and the blue parts are substrates. The unit cell size is 16.5 mm × 16.5 mm (0.25$\lambda_{L}$ × 0.25$\lambda_{L}$), where $\lambda_{L}$ is the free space wavelength at the lowest operating frequency, 4.58 GHz. It is composed of three metal layers separated by two substrate layers; each layer is connected by four coaxial via holes. The thickness of PCM is 4.105 mm, about 0.062$\lambda_{L}$. Additionally, four identical metal patches on the top layer are rectangular patches with internal grooves and added branches, which are placed in an orthogonal clockwise rotation along the periodic edge of the element. Furthermore, four coaxial via holes with the diameter $d_{1}$ = 0.4 mm are connected to the upper and lower four patches, respectively, to form the structure of “receiving antenna-non-radiating coupler-radiating antenna”. The polarizations of the receiving antenna and the radiating antenna are orthogonal. The metal plate in the middle is the common ground of the upper and lower structures. Table 1 and Table 2 show the simulated material parameters of copper and F4B.

Full wave simulation is carried out by Ansoft High Frequency Structure Simulator (HFSS). Figure 2a shows the cross-polarization and co-polarization coefficients of the proposed PCM, where $t_{yx}$ and $t_{xy}$ are the cross-polarization transmission coefficients under x-polarized and y-polarized incident waves, and $r_{xx}$ and $r_{yy}$ are the co-polarization reflection coefficients under x-polarized and y-polarized incident waves, respectively. The results show that the proposed PCM can achieve cross-polarization conversion in the dual bands of 4.58–4.69 GHz and 5.33–5.42 GHz, and the polarization conversion rate (PCR) in the two passbands is greater than 90%. For the x-polarized incident wave, PCR can be calculated as:(1)$$PCR=\frac{{\left|t_{yx}\right|}^{2}}{{\left|t_{yx}\right|}^{2}+{\left|t_{xx}\right|}^{2}}$$

The txy and tyx are −0.22 dB and −0.35 dB, respectively, at two central operating frequencies of 4.64 GHz and 5.37 GHz, and the PCR is up to 95%. The results in Figure 2b verify the polarization omnidirectional characteristics of the proposed PCM. Since the response of the PCM is consistent under x-polarized and y-polarized waves, only the variation of φ from 0° to 45° under x-polarized waves is sufficient to demonstrate its performance.

The simplified circuit model in Figure 3a is obtained by the circuit equivalence of the PCM unit, where the characteristic impedance of free space is denoted by $Z_{0}$ = 377 Ω, and the circuit equivalent of the original model is marked in Figure 1. The parameter values in the equivalent circuit are extracted by using the optimization algorithm in Microwave Office 13 software to fit the EM simulation curves from the HFSS 15 software. The boundary conditions are master and slave in the x and y-axis direction. The maximum edge length of the numerical grid is 3.121 mm, while the minimum size of the numerical grid is 0.035 mm. The distance of the ports from the structure is 15 mm (1/4 $\lambda_{c}$), where $\lambda_{c}$ is the wavelength at the central frequency. The resonators with *Y_au_* and *Y_bu_* admittance are equivalent to the upper inner and outer metal patches, and the resonators with *Y_al_* and *Y_bl_* admittance are equivalent to the lower inner and outer patches, respectively. *Y_au_* can be expressed as:(2)$$Y_{au}=1/Z_{au}=1/\left[\frac{1}{j\omega C_{1}}∥(j\omega L_{1}+\frac{1}{j\omega C_{d}})∥\frac{1}{R_{1}}\right]=Y_{al}$$
where *ω* is the angular frequency, and *Y_bu_* is calculated similarly to *Y_au_*. The equivalent circuit model of the PCM is symmetric with respect to the ground, since the lower metal patches are the same as the upper ones, only at different locations. The coaxial-via-hole connection can be equivalent to the inductance transmission (*L_v_*_1_ and *L_v_*_2_), and the upper and lower resonators are connected in series. The transmission of the inner and outer patches from one end of the PCM to the other can be equivalent to two transmission paths. The transmission matrix of one of the paths can be written as [24]:(3)$$A_{Ι}=\left(\begin{array}{cc} A_{1} & B_{1}\\ C_{1} & D_{1} \end{array}\right)=\left(\begin{array}{cc} 1 & 0\\ Y_{au} & 1 \end{array}\right)\left(\begin{array}{cc} 1 & j\omega L_{v1}\\ 0 & 1 \end{array}\right)\left(\begin{array}{cc} 1 & 0\\ Y_{al} & 1 \end{array}\right)$$

The equation of the other transmission matrix $A_{II}$ is similar to (3). What is more, the inner or outer upper and lower resonators are connected in series, respectively, and then parallel to each other. Here, both $A_{I}$ and $A_{II}$ need to be converted into admittance matrices ($Y=[Y_{11},Y_{12};Y_{21},Y_{22}]$):(4)$$Y_{11}=\frac{D}{B},Y_{12}=\frac{BC-AD}{B},Y_{21}=\frac{-1}{B},Y_{22}=\frac{A}{B}$$

Therefore, the admittance matrix *Y* = *Y_I_* + *Y_II_* of the equivalent circuit in Figure 3a can be obtained, and the cross-polarization transmission coefficient and co-polarization reflection coefficient can be further calculated as:(5)$$t_{yx}/t_{xy}=\frac{-2Y_{21}Y_{0}}{\Delta Y}$$
(6)$$r_{xx}/r_{yy}=\frac{\left(Y_{0}-Y_{11}\right)\left(Y_{0}+Y_{22}\right)+Y_{12}Y_{21}}{\Delta Y}$$
where $Y_{0}=1/Z_{0}$, $\Delta Y=\left(Y_{11}+Y_{0}\right)\left(Y_{22}+Y_{0}\right)-Y_{12}Y_{21}$. Therefore, the equivalent circuit calculation results in Figure 4b are obtained, which are highly consistent with the EM simulation results.

### 2.2. Mechanism of PCM

The dual frequency is due to the different sizes of the inner and outer patches of the patch antenna, which can produce two resonant frequencies. Figure 4a,b demonstrates the electric field intensity distribution on the upper and lower surfaces at the two center operating frequencies of 4.64 GHz and 5.37 GHz under the excitation of the y-polarized wave, respectively. The results show that the outer metal patch resonates at 4.64 GHz, while the inner rectangular patch resonates at 5.37 GHz.

It is known that the electric field ${\vec{E}}_{u}$ of a linearly polarized electromagnetic wave, with arbitrary polarization azimuth *φ*, can be decomposed into components ${\vec{E}}_{x}$ and ${\vec{E}}_{y}$ in the two orthogonal direction x- and y-axes. The proposed PCM exploits this property to achieve omnidirectional polarization conversion. The PCM rotates the two orthogonal components of incident ${\vec{E}}_{u}$ by 90°, respectively. Thus, the ${\vec{E}}_{v}$ at the transmission terminal synthesized from the two new orthogonal components is rotated by 90° with respect to the original ${\vec{E}}_{u}$. This phenomenon is illustrated by the surface current vector distribution at the top and bottom of the unit cell. Taking 4.64 GHz as an example, Figure 5 shows the surface current vector distribution between the top and bottom of the unit at 4.64 GHz when the *φ* of incident wave takes different values.

Take *φ* = 30° as an example, in Figure 5b, where ${\vec{E}}_{u}$ is mainly ${\vec{E}}_{y}$ component and some ${\vec{E}}_{x}$ component exist. After the electromagnetic wave reaches the surface of the PCM, the metal patches placed in the x- and y-directions generate the induced currents of the ${\vec{E}}_{x}$ and ${\vec{E}}_{y}$ components, respectively. The current flows through the coaxial via holes and reaches the bottom, producing a strong current distribution on the patch placed orthogonal to each other. Now, the ${\vec{E}}_{x}$ component of the incident wave becomes ${\vec{E}}_{y}$, and ${\vec{E}}_{y}$ converts to ${\vec{E}}_{x}$. Therefore, the electric field ${\vec{E}}_{v}$ synthesized at the terminal is orthogonal to that of ${\vec{E}}_{u}$, which completes the cross-polarization conversion of the linearly polarized incident waves at any polarization azimuth. Moreover, the current intensity is not diminished before and after the polarization conversion. This indicates that the PCM suffers little energy loss during transmission, since the current induced in the x- and y-directions are independent of each other, and the transmission mode of the coaxial via hole also minimizes the loss.

## 3. Fabrication and Measurement

The proposed PCM prototype is manufactured, as shown in Figure 6a. The measured sample is composed of 12 × 12 periodic arrays. A pair of the identical linearly polarized horn antennas (LB-187-15-C-SF) working at 3.95–5.85 GHz are used to transmit and receive electromagnetic waves, and they are connected to two ports of the vector network analyzer (Anritsu MS46322A, Kanagawa, Japan). The distance between the sample and the horn antenna satisfies the far-field condition. The measured environment is shown in Figure 6b, and the measurement is carried out in the microwave anechoic chamber.

The measured results are normalized by an equal-sized metallic plate. The co-polarization reflection coefficient and cross-polarization transmission coefficient, obtained from the measurement and simulation, are shown in Figure 7. As can be seen from the comparison of simulated and measured results, the tyx curves were in good agreement. However, compared with the simulation result, the measurement result of rxx shifted to the right by about 0.1 GHz at the low resonant frequency, which may be caused by the slight oblique incidence angle of the incident wave due to the relative position of the two horns on the same side. Since the current path at low frequencies is longer than that at high frequencies, low frequency is more sensitive to oblique incidence. In addition, the effect of the incident wave on the reflected wave can also bias the measured results.

The omnidirectional polarization is verified by measurement with different polarization azimuthal angles. On the basis of the above measurement, a certain angle is rotated along the normal line of the horn antenna aperture, and φ = 0°, φ = 15°, φ = 30°, and φ = 45° are successively measured. The measured results for tyx and rxx are shown in Figure 8. According to the measured results, when the polarization azimuth φ varied from 0° to 45°, it had no effect on tyx and rxx. The measured results for tyx and rxx well demonstrate the dual-frequency and omnidirectional polarization characteristics of the proposed PCM.

Comparisons with other reported metasurface-based polarization converters are included in Table 3. It can be seen from Table 1 that the PCM proposed in this communication not only has the advantage of low profile in structure, but also has a low insertion loss in two operating frequency bands. Moreover, it is also characterized by omnidirectional polarization, while the existing polarization converters, 9, 10, 17, 19, and 20, can only work with single or dual polarization.

## 4. Conclusions

In this work, a dual-band transmissive PCM, with both omnidirectional polarization and a low profile, is proposed. The principle of vector orthogonal decomposition is used to achieve the cross-polarization conversion of an arbitrary azimuth linear polarization wave incident. Furthermore, the dual-band operation makes the PCM more practical. Moreover, the PCM realizes the polarization conversion at the level of the planar circuit, which further reduces the thickness of the substrate and makes it ultra-thin. The transmission of electromagnetic waves between the receiving and the radiating patches is equivalent to the coaxial line transmission of electromagnetic waves in the TEM mode, and the currents induced on each patch are independent of each other, which keeps the insertion loss at a low value.

## Figures and Tables

**Figure 1 materials-16-04347-f001:**
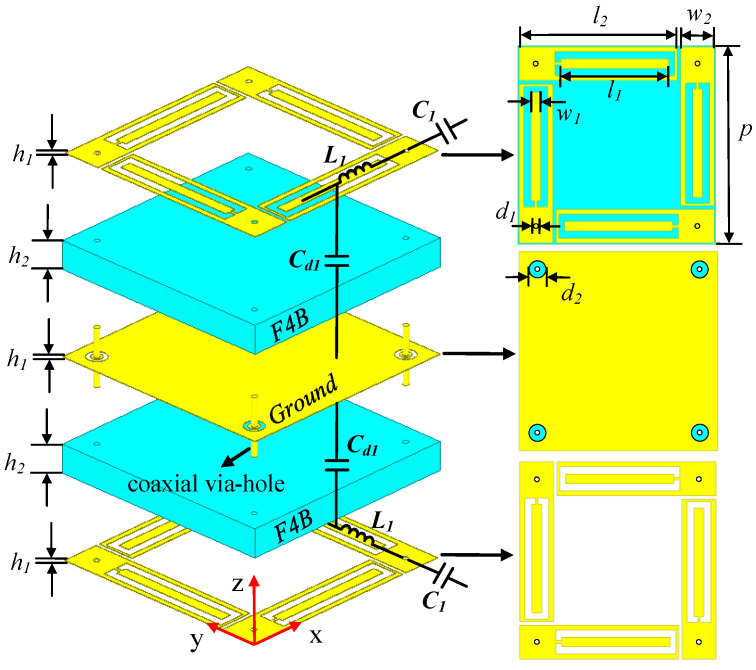
The unit cell of the proposed dual-band PCM where the top layer is connected with the bottom layer by four coaxial via-holes. The geometric dimensions are as follows: $h_{1}$ = 0.035, $h_{2}$ = 2, p = 16.5, $l_{1}$ = 9, $w_{1}$ = 0.8, $l_{2}$ = 13.2, $w_{2}$ = 2.8, and $d_{1}$ = 0.4, $d_{2}$ = 1.4 (unit: mm).

**Figure 2 materials-16-04347-f002:**
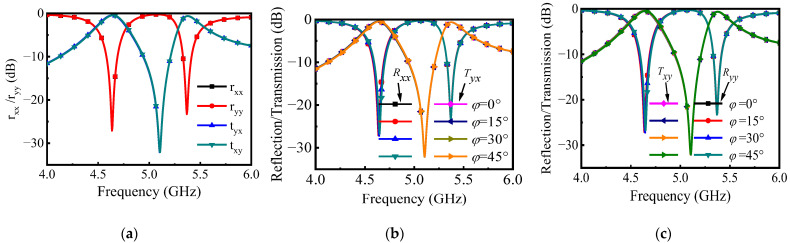
Cross-polarization transmission coefficient and co-polarization reflection coefficient: (**a**) under x-polarized and y-polarized waves; (**b**) when polarization azimuth φ varies from 0° to 45° under x-polarized wave; and (**c**) when polarization azimuth φ varies from 0° to 45° under y-polarized wave.

**Figure 3 materials-16-04347-f003:**
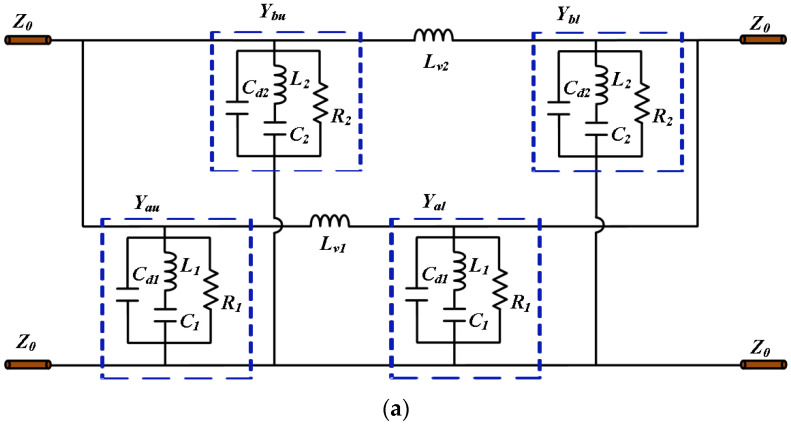
(**a**) Equivalent-circuit model of the proposed PCM ($C_{d1}$ = 9.55 pF, $L_{1}$ = 0.4419 nH, $C_{1}$ = 2.5324 pF, $R_{1}$ = 2772 Ω, ${Lv}_{1}$ = 0.16 nH, $C_{d2}$ = 8.71 pF, $L_{2}$ = 0.6424 nH, $C_{2}$ = 2.3596 pF, $R_{2}$ = 3030 Ω, $L_{v2}$ = 0.38 nH). (**b**) Cross-polarization transmission coefficient and co-polarization reflection coefficient of the PCM are calculated by full-wave simulation and equivalent circuit model.

**Figure 4 materials-16-04347-f004:**
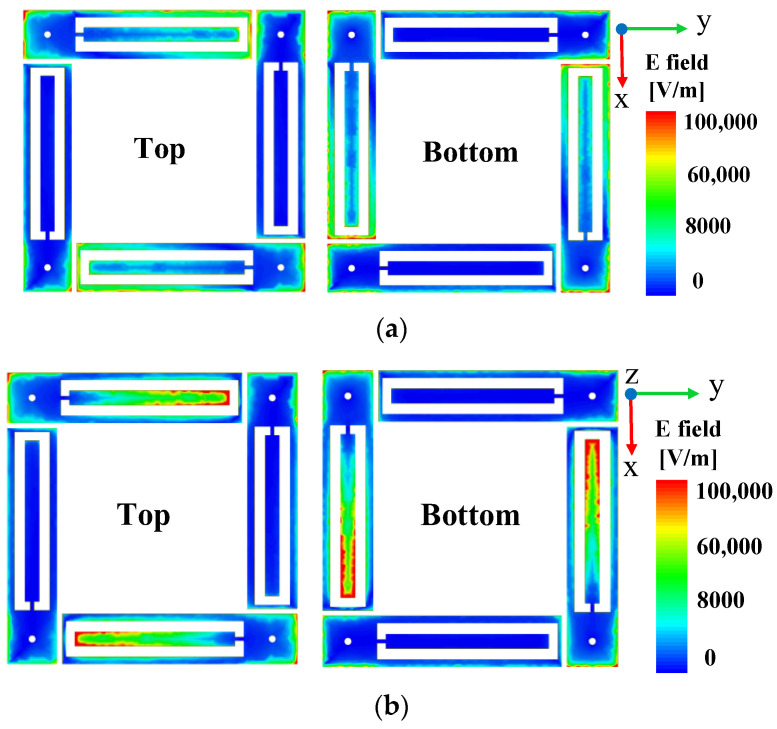
Electric field distribution on the top and bottom surfaces at (**a**) 4.64 GHz and (**b**) 5.37 GHz with y-polarized wave incident.

**Figure 5 materials-16-04347-f005:**
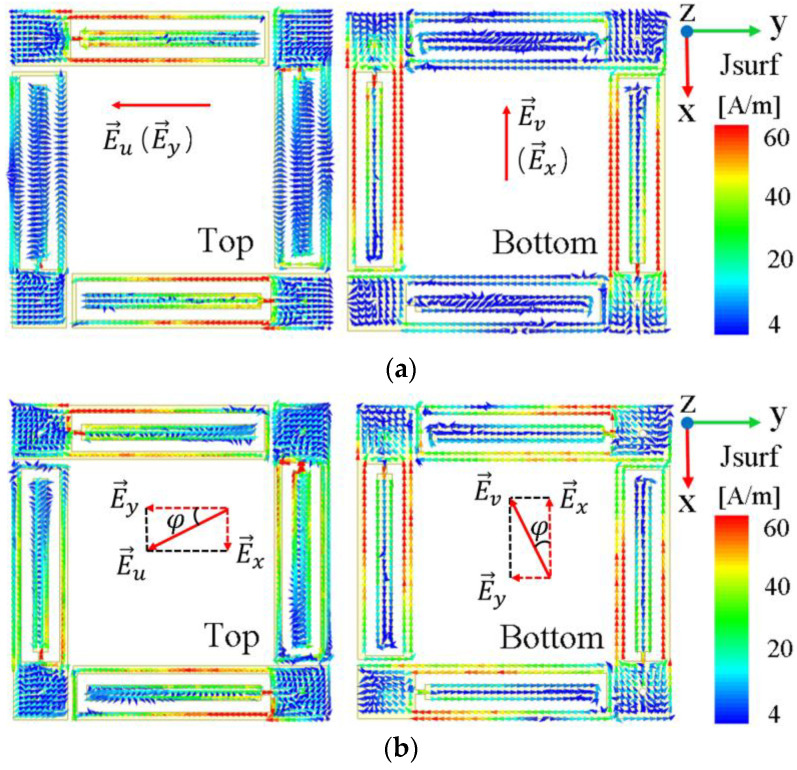
Surface current vector distribution of unit top and bottom at 4.64 GHz when (**a**) φ = 0° and (**b**) φ = 30°.

**Figure 6 materials-16-04347-f006:**
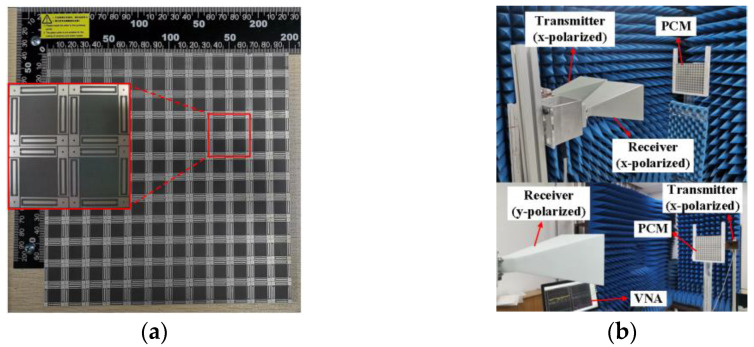
The PCM prototype and experimental set-up. (**a**) Top view of the PCM. (**b**) Experimental setup for co-polarization reflection coefficient and cross-polarization transmission coefficient.

**Figure 7 materials-16-04347-f007:**
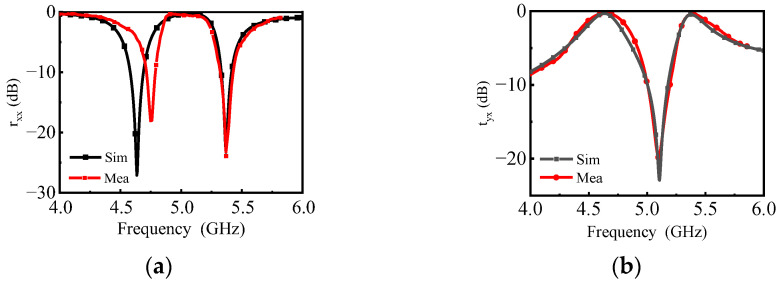
Simulated and measured (**a**) rxx and (**b**) tyx.

**Figure 8 materials-16-04347-f008:**
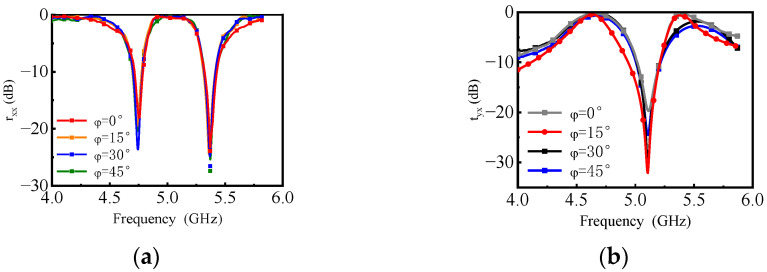
Measured (**a**) rxx and (**b**) tyx of the PCM at φ = 0°, 15°, 30°, and 45°.

**Table 1 materials-16-04347-t001:** Material parameters of copper.

Electric Cond.	Thermal Cond.
$5.96\times 10^{7}$ [S/m]	401 [W/K/m]

**Table 2 materials-16-04347-t002:** Material parameters of F4B.

Relative Permittivity	Loss Tangent
2.65	0.002

**Table 3 materials-16-04347-t003:** Comparison with other works in terms of performance.

Ref.	Passband	PCR	Thickness (*λ_L_*)	Insertion Loss (dB)	* OP
[14]	Dual	90%	0.13	0.37/0.2	No (SP)
[15]	Single	90%	0.0235	1.16	No (SP)
[22]	Single	90%	0.07	0.8	No (DP)
[25]	Dual	88%	0.12	1.94/1.2	No (DP)
[26]	Dual	86%	0.069	0.4/1.9	No (DP)
This work	Dual	90%	0.062	0.22/0.35	Yes

* OP: omnidirectional polarization; SP: single polarization; DP: dual polarization.

## Data Availability

Not applicable.

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
