# Peer review of "Low Profile Dual-Band Polarization Conversion Metasurface with Omnidirectional Polarization"

_materials, 2023, doi:10.3390/ma16124347_

Round 1

Reviewer 1 Report

1. "The upper and lower metal patch layers of the
metasurface are receiving antennas and transmitting antennas, respectively". Please elaborate this line in the abstract, it looks confusing.

2. There is a lot of previous work done in designing of wideband PCM. Then, please explain significance of the proposed dual narrow band PCM.

none

Author Response

    The revised manuscript has greatly benefited from the constructive comments and valuable suggestions. We would like to thank you sincerely for the time and effort spent to help improve the presentation of this paper.Detailed response are in Word file.

Reviewer 2 Report

The paper is well-written, presenting a thin metasurface for converting one polarization to the other in incident EM waves, over two frequency bands. The authors report excellent agreement between simulations and experiments for the above functionality, as well as its physical justification, making one confident that the presented results are sound.

The presented topic (metasurfaces + polarization conversion) is also timely. As a result of all these, I feel that the paper could potentially be accepted for publication but only if (from my perspective) the authors satisfactorily address the following points, which would substantially improve its quality:

1. The most important question in over how large span of angles is this functionality attained over both bands? Does it only work for perpendicular incidence? If yes, coud it also work for other angles?

2. A band-diagram of the reported periodic structure for both incident polarizations would be helpful for clearly seeing the operational bands and cross-polarization performance.

3. Can the proposed design / methodology be also implemented on-chip? - see, e.g.,  ACS Photon. 5, 301–305 (2018).

4. Can the operational bandwidths (which are rather narrow) and the transmission efficiency (which is a key figure-of-merit) be further increased / improved?

5. Important, relevant previous works (of interest to prospective readers), such as J. Opt. Soc. Am. B 38, C50-C57 (2021)J. Opt. 49, 17–22 (2020), etc (as well as the work cited in Q. 2 above), are currently missing, and should be cited in a revised version of the work.

Author Response

The revised manuscript has greatly benefited from the constructive comments and valuable suggestions. We would like to thank you sincerely for the time and effort spent to help improve the presentation of this paper.Detailed response are in the Word file.

Reviewer 3 Report

This paper describes a thin polarization converter based on metasurface that operates in dual microwave frequency and any polarization direction. The authors demonstrated that it offers high polarization conversion efficiency and low insertion loss both numerically and experimentally. It would be of interest to Meterials readers, however, I have some concerns as follows:

1) Fig. 2. The authors should indicate how they determine the solid lines (the equation and parameters of the curve), since the values around the resonant frequency are critical.

2) Page 6, last paragraph. The authors should mention why the slight obliqueness of the incident wave affects the low resonant frequency of rxx at 4.64 GHz but not the high resonant frequency at 5.37 GHz in Fig. 7(a).

Followings are minor deficiencies:

1) Abstract, line 8: “λL” should read “λL”. (The “L” is a subscript.)

2) Introduction, 1st paragraph, line 3 (and same for others): Reference number “1, 2” should be indicated as “[1, 2]”. (Square brackets are necessary.)

3) Page 2, Section 2, 1st paragraph. “λl” should read “λL”. (It should be identical throughout the text whether the “L” is an upper or lower case letter.)

4) Page 2, last paragraph, line 2 from the bottom (and same for others): “tyx and txy” should read “tyx and txy”. (The “yx” and “xy” are subscripts.)

5) Fig. 2. The plots for rxx and txy cannot be recognized.

6) The lower limit of vertical axis in Fig. 2 (a) should be matched with that in Fig. (b).

7) Page 4, 1st line below Eq. 2 (and same for Eq. 6): It should not be indented.

8) Page 7, 1st line above Table 1. “(λL)” seems to be unnecessary.

9) Table 1. The authors should describe the meaning of “OP”, “SP” and “DP”.

10) Ref. 11. Page number is missing.

Author Response

(The authors gave the same response as above.)

Reviewer 4 Report

The authors have proposed low profile dual-band polarization conversion metasurface with omnidirectional polarization. The transmission and reflection characteristics have been calculated by numerical simulation and basic properties have been shown. In addition, the results of experimental measurement agree well with numerical results. Therefore, I think this paper to be published.

Author Response

(The authors gave the same response as above.)

Reviewer 5 Report

The authors proposed a dual-band transmissive polarization conversion metasurface with both omnidirectional polarization and low profile. Importantly, both a theoretical model of the device and its experimental prototype were developed. As a result of careful evaluation, I believe that the current version of the manuscript lacks a more detailed reference to the following:

1.      The introduction lacks justification for the use of metamaterials in polarization converters. Why did the authors decide to use a metamaterial? What are the advantages of polarity converters based on metamaterials compared to other concepts of this type of devices? In this context, it is worth mentioning a wide range of metamaterials-based photonic devices: Scientific Reports 9.1 (2019): 20367; Sensors 18.12 (2018): 4209; Applied Physics Letters 108.22 (2016): 224101; etc.

2.      The metasurface scheme is quite unusual. Where did the concept come from? What is the unique feature of the proposed metasurface from the point of view of its use in polarization converters?

3.      For better clarity please insert a table with exact material parameters (permeability, loss) of all materials that were used to simulate the theoretical model.

4.      Did the authors take into account the dispersion of material parameters in the simulations? What were the boundary conditions in the simulations? What was the size of the numerical grid? What was the distance of the ports from the structure? This and other information must be completed.

5.      Were formulas (2)-(6) first developed by the authors? If not, please indicate the source.

6.      The authors have included a table comparing the parameters of the developed device to others previously developed by other authors. Given that, what are the unique uses of the metadevice?

In some parts of the manuscript English is rather poor.

Author Response

(The authors gave the same response as above.)

Round 2

Reviewer 1 Report

The manuscript can be accepted in its present form.

Reviewer 2 Report

I have read the revised version of the work, and the authors' replies to my previous comments. The authors have appropriately amended and updated the main manuscript (responding to the comments received by all referees) by including additional results and clarifications, and have reasonably well addressed my previous comments to them. As such, I feel that this revised and improved manuscript is now ready to appear in the journal as is.

Reviewer 5 Report

The authors made corrections to the manuscript in line with my comments. Therefore, I believe that the manuscript should be published in its present form.